# Systematic Design of Polypyrrole/Carbon Fiber Electrodes for Efficient Flexible Fiber-Type Solid-State Supercapacitors

**DOI:** 10.3390/nano10020248

**Published:** 2020-01-30

**Authors:** Yu-Shun Sung, Lu-Yin Lin

**Affiliations:** 1Department of Chemical Engineering and Biotechnology, National Taipei University of Technology, Taipei 10608, Taiwan; zxc841226@gmail.com; 2Research Center of Energy Conservation for New Generation of Residential, Commercial, and Industrial Sectors, Taipei 10608, Taiwan

**Keywords:** carbon fiber, electropolymerization, flexible, polypyrrole, supercapacitor

## Abstract

Fiber-type supercapacitors (FSC) have attracted much attention as efficient energy storage devices for soft electronics. This study proposes the synthesis of polypyrrole (PPy) on carbon fiber (CF) using electropolymerization as the energy storage electrode for FSC. Effects of the electrolyte, applied current, and time of electropolymerization for synthesizing PPy on CF are investigated. The configuration of the electrochemical system is also studied to better understand the electropolymerization of PPy. The highest specific capacitance (C_M_) of 308.2 F/g are obtained for the PPy electrode prepared using 0.5 M pyrrole and 0.3 M NaClO_4_ as the electrolyte at 40 mA for 20 min. The FSC assembled with PPy electrodes and the polyvinyl alcohol/H_3_PO_4_ gel electrolyte shows a C_M_ value of 30 F/g and the energy density of 5.87 Wh/kg at the power density of 60.0 W/kg. Excellent cycling stability with C_M_ retention of 70% and Coulombic efficiency higher than 98% in 3000 times charge/discharge process, and the good bending capability with C_M_ retention of 153% and 148%, respectively, under the bending angle of 180° and the bending times of 600 are achieved. This work gives deeper understanding of electropolymerization and provides recipes for fabricating an efficient PPy electrode for soft energy storage devices.

## 1. Introduction

To achieve a more convenient and comfortable life, human beings have started to develop soft electronics for monitoring physiological index, controlling environment temperature, and generating fancy displays on human bodies in recent years [1,2,3]. To supply energy for operating soft electronics, the energy generation and storage devices are of the great importance to be developed. The supercapacitor with high power density and long cycle life has been studied for many years [4,5,6]. Rani et al. introduced thiol-functionalized, nitrogen-doped, reduced graphene oxide scrolls as the electrode materials for an electric double-layer supercapacitor assembled in a coin cell and achieved high energy and power densities for the device [7]. However, to fit the feature of the soft electronics, the flexible substrate is required to construct the SC [8,9,10,11,12,13,14,15,16]. The fiber-type solid-state supercapacitor (FSC) is more promising to combine with soft electronics for developing the fiber-to-fiber device with better contact. In addition, it is much easier to weave the fiber-type device in the cloth to realize the zero distance between the human body and soft electronics. 

The FSC has been reported for many years [17,18,19,20]. The active material of FSC is similar to those of the normal type SC, such as carbon materials [21,22,23], conducting polymers [24,25,26], and metal compounds [27,28,29]. The conducting polymer with the highly bendable and stretchable features has been considered as the most promising active material for FSC [30,31]. Chen et al. fabricated the polypyrrole (PPy)/graphene composite on the carbon nanofiber yarn as the energy storage electrode of FSC with the polyvinyl alcohol (PVA)/H_3_PO_4_ gel electrolyte. A specific capacitance (C_L_) of 111.46 mF/cm at 2 mV/s was obtained [32]. Guo et al. fabricated carbon nanotube/PPy composites using the electrochemical deposition technique as the active material for FSC [33]. Sun et al. used urethane yarns as the wearable scaffold for hosting carbon nanotubes and PPy to fabricate stretchable yarn electrodes for FSC [34]. Abdah et al. fabricated the supercapacitor electrode with PPy coated on carbon fiber (CF) using electrospinning, carbonization, and in-situ polymerization techniques [35]. The carbon materials are commonly applied to combine with the conducting polymer as the active material for the FSC in the previous literatures. However, the systematic study on the physical and the electrochemical properties of the bare conducting polymer relating to the synthesizing parameters is limited. It is expected that the bare conducting polymer can still present the excellent electrochemical performance without other active materials incorporated once the suitable synthesizing parameters were applied for fabricating the conducting polymer electrodes. 

In this work, PPy was coated on CF (CF@PPy) by using the electropolymerization technique and applied as the energy storage electrode for FSC. The PPy has been well applied as the efficient active material for SC, due to its high conductivity, facile redox activity, processability, and low environmental impact features [36,37,38,39]. The synthesizing parameters such as electrolyte concentration, participation of the supporting electrolyte, the electropolymerization applied current, and time, as well as the electropolymerization system design were carefully investigated to achieve assemble the efficient FSC with excellent energy storage electrodes based on PPy. The supporting electrolyte is found to be necessary for achieving the desired applied current for electropolymerization. The larger applied current and longer time for electropolymerization lead to the larger thickness of PPy deposited on CF. The highest specific capacitance (C_M_) of 308.2 F/g at 5 mV/s was obtained for the CF@PPy electrode prepared using the electrolyte containing 0.5 M pyrrole and the NaClO_4_ supporting electrolyte with the applied current of 40 mA and the deposition time of 20 min. The FSC was assembled using the two CF@PPy electrodes and the PVA/H_3_PO_4_ gel electrolyte. The potential window of 1.2 V and the C_M_ value of 30 F/g at 0.1 A/g were obtained for the device. The C_M_ retention of 70% and the Coulombic efficiency higher than 98% were also attained in 3000 times repeated charge/discharge process. The bending test was also carried out with the C_M_ retention of 153% and 148% under the bending angle of 180° and the bending time of 600, respectively. This work provides a simple and time-saving method to construct efficient conducting polymer-based FSC. It is expected that the smart cloth can be developed by weaving the FSC with normal cotton threads and applied on soft electronics effectively. 

## 2. Experiments 

### 2.1. Fabrication of Polypyrrole on Carbon Fiber (CF@PPy)

The CF (TC-33, 3k, Formosa Plastic Group) was cleaned using the neutral clearer, deionized water (DIW), acetone (uni-onward), and acetonitrile (99.9%, uni-onward), sequentially. The CF was coated by pyrrole (99%, Acros) using the electropolymerization technique in a three-electrode system, where the cleaned CF is the working electrode, a Pt wire is the counter electrode, and an Ag/AgCl electrode is the reference electrode. The electrolyte contains different concentrations of the pyrrole monomer and the supporting electrolyte (NaClO_4_, 98–102%, Alfa Aesar). Different currents and times were applied for coating different amounts of PPy on CF at room temperature. The electrolyte was replaced with the fresh one after each coating of pyrrole on CF using electropolymerization. After carrying out the electropolymerization, the electrode was rinsed using alcohol (EtOH, 95%, uni-onward) and dried at 60 °C overnight. 

### 2.2. Assembly of Flexible Fiber-Type Solid-State Supercapacitors

The FSC was assembled using the CF@PPy electrode and the CF electrode. Each of the electrode was dipped in the gel electrolyte containing 5 g PVA (98–99%, Alfa Aesar, MA, USA) and 5 mL H_3_PO_4_ (analytical reagent grade, fisher, MA, USA) in 50 mL DIW for 20 min before being twisted on each other. After fully covering the gel electrolyte on the electrode, two CF@PPy electrodes were twisted on each other, and then the FSC was dipped in the gel electrolyte again for fully covering the electrolyte on the device. Afterward, the FSC was dried at the room temperature for 15 min, and then the FSC was obtained. 

### 2.3. Measurements and Characterizations

Morphology of PPy was examined using the field-emission scanning electron microscopy (FE-SEM, Nova NanoSEM 230, FEI, Hillsboro, OR, USA). The composition of CF@PPy was examined using the X-ray diffraction patterns (XRD, X’Pert^3^ Powder, PANalytical) and the Raman spectrometer (Dongwoo, DM500i) with the green laser excitation (514.5 nm). The electrochemical performances of the CF@PPy electrodes and the FSC were evaluated using the cyclic voltammetry (CV), Galvanostatic charge/discharge (GC/D), and the electrochemical impedance spectroscopy (EIS) techniques, which were carried out by using the potentiostat/galvanostat instrument equipped with an FRA2 module (PGSTAT 204, Autolab, Eco–Chemie, the Netherlands). The open-circuit potential and the frequency ranges from 0.01 Hz to 100 kHz were used to carry out the EIS measurement. 

## 3. Result and Discussion

### 3.1. Effects of Electropolymerized Electrolyte on Electrochemical Performance of CF@PPy 

The CF was coated by pyrrole using the electropolymerization technique. The pyrrole monomer can be oxidatively polymerized to be PPy using chemical oxidation process or electropolymerization process. In a liquid medium and with certain electric field used in this work, the pyrrole monomer can be deposited on CF using electropolymerization. The supporting electrolyte of NaClO_4_ is used to enhance the ionic strength and conductivity of the solution by eliminating the IR drop. Hence even the diffusion coefficient of pyrrole remains almost constant the enhancement on the solution conductivity can improve the deposition of pyrrole with supporting electrolyte. The parameters for electropolymerization such as electrolyte, applied current, and applied time are important for deciding the morphology and electrical conductivity of PPy and the electrochemical performance of the CF@PPy electrodes. Firstly, the composition of the electrolyte was studied, including the pyrrole monomer concentration and the participation of the supporting electrolyte, NaClO_4_. Appendix A shows the CV curves for the CF@PPy electrodes prepared using the electropolymerization electrolyte of 0.5 and 1.0 M pyrrole monomers. The curve for the pure CF without coating PPy was also shown in this figure. The larger integrated area of the CV curves was obtained for the CF@PPy electrodes comparing to that for the pure CF electrode. However, the enhancement on the current is not large. It is inferred that the pyrrole monomers may not be able to fully diffuse and deposit on CF with these concentrations. Therefore, 0.3 M NaClO_4_ was added in the electrolyte as the supporting electrolyte for enhancing the diffusion of the ions. Since the supporting electrolyte may be helpful for ions diffusion, the higher pyrrole concentration of 1.5 M was also used for carrying out the electropolymerization. The integrated area of the CV curves was enhanced largely when the supporting electrolyte was used for the electropolymerization, owing to the increased ion diffusion for coating sufficient PPy on CF for energy storage. Appendix A lists the specific capacitance (C_L_) for the CF@PPy electrodes prepared using different electrolytes for electropolymerization. The C_L_ value was calculated based on Equation (1).
(1)$$C_{L}\;=\;\int \mathrm{IdV}\;/\;(v\;L)$$
where ∫IdV is the integrated area of the CV curve, *v* is the scan rate for measuring the CV curve, and L is the working length of the electrode. The pure CF was indicated as 0.0 M PPy in this table. More than ten-fold enhancement on the C_L_ value was obtained when the supporting electrolyte was used for electropolymerization. The optimized CF@PPy electrode prepared using 1.0 M pyrrole and 0.3 M NaClO_4_ presents the largest C_L_ value of 93.75 mF/cm, which is higher than 200-fold of the C_L_ value for the pure CF electrode (0.44 mF/cm). However, since the C_L_ value difference between the CF@PPy electrodes prepared with the supporting electrolyte is not large, the smallest concentration of pyrrole (0.5 M) was considered to be the optimized composition of electrolyte in considering of both the specific capacitance and the cost of the pyrrole monomers. Further investigations on the applied current and applied time for electropolymerization were then discussed. 

### 3.2. Effect of the Applied Current for Electropolymerization on Physical and Electrochemical Properties of CF@PPy

After optimizing the electrolyte composition and proving the importance of the supporting electrolyte, the effects of the applied current for electropolymerization on the morphology of PPy and the electrochemical properties of CF@PPy electrodes were further studied. Figure 1(a,f), (b,g), (c,h), (d,i), and (e,j) respectively shows the SEM images of the CF@PPy prepared using the applied current of 10, 20, 30, 40, and 50 mA. Since the electropolymerization time is the same, the higher applied current is expected to provide more holes for electropolymerization. As expected, the thickness of PPy increases for the CF@PPy electrodes prepared with higher applied currents. However, the thickness is not fully proportional to the currents applied for electropolymerization. The PPy thin film was formed on CF when the smallest current of 10 mA was applied for electropolymerization, whereas several particle-like structures were observed for the PPy prepared using the higher applied currents of 20 and 30 mA. Furthermore, some particle-like structures disappeared for the CF@PPy electrodes prepared using higher applied currents of 40 and 50 mA. These phenomena suggest that the PPy can be electropolymerized on CF evenly when small currents were applied, but with the higher applied currents the uneven surface of PPy would be formed on CF due to the randomly distributed charges. That is, when the electropolymerization current is larger, the time for arranging charges becomes shorter. The short time for the charge arrangement would lead to the uneven distribution of PPy on CF. On the other hand, the disappearance of the particle-like structure for the CF@PPy prepared using the largest applied currents is probably owing to the higher charge density in the gaps between the particle-like structures. Hence, the more PPy would deposit in the gaps between the particle-like structures when higher currents (40 and 50 mA) were applied for synthesizing CF@PPy. This process would lead to the filling of the gaps and to the flatter surface of CF@PPy. Appendix A shows the chronopotentiometric curves for the electrodes prepared using 10, 20, 30, 40, and 50 mA. The larger potential was obtained when larger current was used for electropolymerization. The theoretical water oxidation potential is around 0.62 V vs. Ag/AgCl in the neutral electrolyte. It is found that the potential is higher than 0.62 V for all the electrodes. Hence it is inferred that the water oxidation process was initiated in the electropolymerization process for all the cases. In addition, by using the larger current for fabricating the electrodes the more oxygen can be produced to influence the growth of PPy more seriously. Furthermore, the electrochemical performance of the CF@PPy electrodes prepared using different applied currents was examined using the CV curves, as shown in Figure 2a. The C_L_ values for the CF@PPy electrodes prepared using different currents were listed in Appendix A for comparison, and the specific capacitances based on the loading mass (C_M_) were also shown in Appendix A for more fair evaluation. One couple of the redox peaks was observed in the CV curves for all the samples, probably attributed from the redox reaction of PPy. The peak separation becomes larger for the CF@PPy electrodes prepared using higher applied currents, indicating the less charge/discharge reversibility for these samples. Since the charge/discharge reversibility of the carbon material is reported to be better than the conducting polymer [40], the worse charge/discharge reversibility for the CF@PPy electrode prepared using higher applied currents could be contributed to the less exposure of the carbon substrate to the electrolyte. In addition, the CV integrated area in proportion to the C_L_ value increases for the CF@PPy electrode prepared using higher applied currents, due to the higher loading mass of PPy for generating more Faradaic reactions with the electrolyte. The highest C_L_ value of 244.9 mF/cm was obtained for the CF@PPy electrode prepared using 40 mA. However, the CF@PPy electrode prepared using 50 mA shows a decreased C_L_ value of 180.9 mF/cm. This phenomenon may be due to the densely packed PPy to expose less surface area to the electrolyte for generating electrochemical reactions. As observed in the SEM images, the CF@PPy electrode prepared using 50 mA (Figure 1e,j) shows smaller diameter than that for the CF@PPy prepared using 40 mA (Figure 1f,i). This phenomenon supports the inference that the PPy is packed too dense for the CF@PPy electrode prepared using 50 mA. On the other hand, the C_M_ values present different trends to the C_L_ values. The C_M_ values are found to increase for the CF@PPy electrodes prepared using smaller applied currents, owing to the smaller loading mass of PPy on CF prepared using smaller applied currents. The highest C_M_ value of 238.1 F/g at 20 mV/s was obtained for the CF@PPy electrode prepared using 20 mA. Since the electrode is based on CF, it is inferred that the lengthy capacitance may be more important than the weighty capacitance. Additionally, the C_M_ value for the CF@PPy electrode prepared using 40 mA is not much smaller than the largest C_M_ value. Therefore, the optimized applied current is inferred to be 40 mA for synthesizing the CF@PPy in this work. Furthermore, the electrochemical performance of the CF@PPy electrodes was also examined using the GC/D curves, as shown in Figure 2b. The specific capacitance is proportional to the discharge time of the GC/D curve. The trend of the specific capacitance is the same as that obtained from the CV curves. The specific capacitance of the CF@PPy electrode increases with increasing electropolymerization current and reaches the maximum value for the sample synthesized using 40 mA. The optimized CF@PPy electrode prepared using 40 mA also shows the smallest IR drop, suggesting the best electrochemical performance for this sample, due to the enough deposition and suitable packing density of PPy to expose large surface area of PPy to the electrolyte. 

The charge transfer resistance was further investigated using the EIS technique. The Nyquist plot for the CF@PPy electrodes prepared using different applied currents was shown in Figure 2c, and Figure 2d presents the equivalent circuit for fitting the charge-transfer resistances. Appendix A lists the series resistance (*R*_S_) and charge transfer resistance at the electrode/electrolyte interface (*R*_ct_) for the CF@PPy electrodes prepared using different applied currents. The *R*_S_ value was decided by resistance in the whole device, including the electrodes and electrolyte. Since the CF@PPy electrode was prepared using different currents, the interaction between the active material and the electrolyte is considered to be varied and hence the *R*_S_ value was varied for the electrodes prepared using different applied currents. Exclusive of the resistances for the CF@PPy electrode prepared using 20 mA, the *R*_S_ and *R*_ct_ values decrease for the electrodes prepared using larger currents. The smallest *R*_S_ and *R*_ct_ values were obtained for the electrode prepared using 40 mA and these values increase again for the electrode prepared using 50 mA. The smallest *R*_S_ and *R*_ct_ values for the CF@PPy electrode prepared using 40 mA is owing to the sufficient deposition and surface exposure of PPy to the electrolyte. The much larger *R*_ct_ value for the CF@PPy electrode prepared using 10 mA comparing to those for the CF@PPy electrodes prepared using higher applied currents is owing to the much poor deposition of PPy on CF that the surface of the CF substrate is even exposed to the electrolyte (Figure 1a,f). It is worthy to discuss the small *R*_S_ and *R*_ct_ values for the CF@PPy electrode prepared using 20 mA. From the SEM image it can be found that the obvious PPy deposition was happened when the current of higher than 20 mA was applied for preparing the electrode. The fully deposition of the PPy for this case may improve the charge transfer and reduce the charge transfer resistance. However, when more PPy was deposited on the CF and several aggregations were formed on the surface (30 mA), the discontinuous film would lead to the enhancement on the charge transfer resistance. This phenomenon vanished for the electrode prepared using 40 mA which shows much uniform PPy film on the CF electrode. 

### 3.3. Effect of Electropolymerization Time on the Physical and Electrochemical Features of CF@PPy

Based on the optimized electrolyte and the electropolymerization current of 40 mA, the effect of the electropolymerization time was further studied. Figure 3(a,d), (b,e) and (c,f) respectively shows the SEM images of the CF@PPy electrodes prepared using the electropolymerization times of 10, 20, and 30 min. Several pores were observed on the surface of the CF@PPy electrode prepared using the smallest time of 10 min, whereas the CF@PPy electrodes prepared using 20 and 30 min show no pores on the surface. Instead, several aggregations were obtained on the surface of CF@PPy prepared using 20 and 30 min. In addition, the diameter of the CF@PPy electrode prepared using 10 min is smaller than those for the CF@PPy electrodes prepared using 20 and 30 min, and the diameters for the CF@PPy electrodes prepared using 20 and 30 min are similar. This phenomenon suggests that PPy was electropolymerized on CF in the vertical direction at the first 10 min and then deposited in the pores with longer electropolymerized times.

The composition of CF@PPy was further analyzed by using XRD patterns, as shown in Figure 3g. The peaks at the 2 theta values of 27° and 45° corresponding to the standard pattern of PPy [41] were observed for all the CF@PPy electrodes, suggesting the successful deposition of PPy on CF using the electropolymerization technique. Due to the similar 2 theta values for the peaks of carbon and PPy [42], it is hard to tell the existence of carbon in the electrode. However, the substrate used for electropolymerization is truly the CF, so verifying the existence of carbon is not as important as that for the existence of PPy. Furthermore, the Raman spectra for the CF@PPy electrodes prepared using different times were shown in Figure 3h. The C-C in-plane deformation at 1310–1400 cm^−1^ and the C-C backbone stretching at 1579 cm^−1^ were observed in all Raman spectra for CF@PPy electrodes [43]. The Raman spectra for all the CF@PPy electrodes are highly similar to the PPy pattern reported in previous works [43,44,45,46], again indicating the successful deposition of PPy on CF using the electropolymerization technique. 

The electrochemical performance of CF@PPy electrodes prepared using different electropolymerization times was further analyzed using CV and GC/D curves, as respectively shown in Figure 4a,b. The corresponding C_L_ values and loading masses were listed in Appendix A for clear comparison. The smaller integrated area of the CV curve and the shorter discharge time of the GC/D curve were obtained for the CF@PPy electrode prepared using 10 min, as compared with those for the CF@PPy electrodes prepared using 20 and 30 min, suggesting the worse electrochemical performance for the former case due to the insufficient deposition and the discontinuous pores on the surface. The C_L_ and C_M_ values are similar for the CF@PPy electrodes prepared using 20 and 30 min. Due to the higher loading mass of PPy prepared using 30 min, the smaller C_M_ value and the larger C_L_ value were obtained for the CF@PPy electrode prepared using 30 min, compared to those for the CF@PPy electrode prepared using 20 min. Since it is better to use shorter time to fabricate electrodes, the optimized electropolymerization time for preparing the CF@PPy electrode is considered to be 20 min in this work. 

The series resistance and charge-transfer resistance were further analyzed using the Nyquist plot, as shown in Figure 4c, and the corresponding equivalent circuit was shown in Figure 4d. The *R*_S_ and *R*_ct_ values were also listed in Appendix A for clear comparison. The much larger resistances were obtained for the CF@PPy electrode prepared using 10 min. This result is consistent with the smallest capacitance for this sample due to the insufficient deposition of PPy on CF. The similar resistance values were obtained for the CF@PPy electrodes prepared using 20 and 30 min, suggesting that the series and charge-transfer resistances play important roles on the capacitance of the electrode. However, although the specific capacitance and the resistances are similar for the CF@PPy electrodes prepared using 20 and 30 min, the high-rate capacitance is varied for these electrodes. Appendix A–c respectively shows the CV curves at different scan rates for the CF@PPy prepared using 10, 20, and 30 min. Appendix A also lists the specific capacitances for the CF@PPy prepared using different times for electropolymerization measured using CV curves at different scan rates. The shape distortion on the CV curve is more serious for the CF@PPy electrodes prepared using longer times. To have more obvious comparison, Figure 4e shows the relation between the scan rate for measuring the CV curves and the corresponding specific capacitance based on the unit length and the unit weight for the CF@PPy electrodes prepared using different times. At smaller scan rates, the C_M_ values are similar for the CF@PPy electrode prepared using different times. However, when larger scan rates were applied for measuring the CV curves, the smaller C_M_ values were obtained for the CF@PPy electrodes prepared using longer times. It is inferred that the thicker PPy layer for the electrode fabricating using longer times may hinder the ion diffusion and limit the utilization of the active material. This phenomenon could be more obvious when larger scan rates were applied for the measurement. Moreover, the C_L_ value is larger for the CF@PPy electrode prepared using longer times, due to the more deposition of PPy and the lack of the loading mass normalization. Similarly, the decrease on the C_L_ value with increasing scan rates is more serious for the CF@PPy electrode prepared using longer times. On the other hand, Appendix A respectively shows the GC/D curves at different current densities for the CF@PPy electrodes prepared using 10, 20, and 30 min. Appendix A also lists the specific capacitances for the CF@PPy electrodes measured using GC/D curves at different current densities. The GC/D curves for all the samples present high symmetry, but the IR drop becomes larger at the high voltage range for the CF@PPy electrode prepared using longer times. The relation between the current density for measuring the GC/D curves and the corresponding specific capacitance based on the unit length and the unit weight for the CF@PPy electrodes prepared using different times were presented in Figure 4f. The reduction on the C_L_ and C_M_ values is more serious for the CF@PPy electrode prepared using longer times. These results indicate the worse high-rate capacity for the electrode with thicker and denser PPy layer on CF, owing to the lack of the full usage of the active material and the inefficient diffusion of ions in the electrolyte.

### 3.4. Effects of Electropolymerization Systems on the Electrochemical Performance of CF@PPy

This study applied the electropolymerization technique to fabricate the CF@PPy electrodes. Usually, the parameters such as electrolyte, applied current, and time are controlled to find the optimized conditions for fabricating the electrodes. However, the design of the electropolymerization system has never been studied. To accelerate the fabrication process, it may be allowed to put more fiber-type substrates in one electrode for electropolymerization. It is worthy to investigate the influence of the fiber assembly in the working electrolyte on the electrochemical performance of the fiber-type electrodes based on the same electropolymerization electrolyte, applied current and time. Herein, three types of the fiber set were assembled in the working electrode for electropolymerization, as shown in Figure 5. The scheme under the CV plot is the corresponding electropolymerization system design. Type A and type B both include three fibers in the working electrode, but the fibers were set in the working electrode in different ways. For type A, three fibers were assembled at different locations with the same distance in-between on one copper tape, which was directly clipped in the working electrode. For type B, three fibers were put together and directly clipped in the working electrode. Type C is the normal one which only includes one fiber in the working electrode. The CV curves of the CF@PPy electrodes prepared using type A, type B, and type C configurations were shown in Figure 5a–c, respectively. For the comparison of the three samples prepared using type A, the integrated area of the CV curve is the largest for sample 1 and the smallest for sample 3. That is, the CF@PPy electrodes prepared at the location closer to the clip shows larger integrated areas and hence larger C_L_ values. For the comparison of the three samples prepared using type B, the CV curves of the three samples are almost overlapped and hence the integrated areas of their CV curves are nearly the same. This phenomenon is reasonable that these three electrodes were put in one clip together without variation on the distance to the clip. Furthermore, it is found that the integrated area sum of the three CV curves for the CF@PPy electrodes prepared using type A is the same as the integrated area sum of the three CV curves for the CF@PPy electrodes prepared using type B. In addition, the integrated area sum for the three CV curves of the CF@PPy electrodes prepared in the type A and type B systems is the same as the CV integrated area of the CF@PPy electrode fabricated using the type C system. The result indicates that by using the same electrolyte, applied current, and time for electropolymerization, the same charge amount could be supplied for electropolymerization regardless of the number of the fiber substrates put in the working electrode and the configuration of the fiber set. However, it may be different on the electrochemical performance when larger currents and times were applied on the electropolymerization, especially when the total charge amount supply is over the limit of one fiber. 

### 3.5. Electrochemical Performance Analysis of the FSC with CF@PPy electrodes

To achieve the real application, the flexible solid-state FSC was assembled using the CF@PPy electrodes. The electrochemical performance of the FSC was firstly analyzed by measuring the CV curve with different potential windows at the scan rate of 2 mV/s, as shown in Figure 6a. A sudden increase on the current of the CV curve was observed at the potential of around 1.4 V, which is attributed to the water oxidation reaction. To avoid the gases generation and to improve the cycling stability of the electrode, the optimized potential window of 1.2 V was determined and used for further measurement. Figure 6b shows the CV curves measured using different scan rates with the potential window of 1.2 V. The shape distortion on the CV curves is limited, suggesting the excellent high-rate charge/discharge capacity for this device. On the other hand, the GC/D curves measured with different potential windows at 0.1 A/g were shown in Figure 6c. The distortion of the GC/D curve occurs with the potential of 1.4 V, suggesting the suitable potential window of 1.2 V for this FSC. Figure 6d presents the GC/D curves at the potential window of 1.2 V measured using different current densities. All the curves show high symmetry regardless of the potential window and the applied current density, again suggesting the excellent high-rate capacity for this FSC. However, the large IR drop was observed in the GC/D curves. The charge-transfer resistances for the FSC were thus examined using the Nyquist plot, as shown in Appendix A. The FSC shows the *R*_S_ and *R*_ct_ values of 13.0 and 14.6 Ω, respectively. The charge-transfer resistances are not extremely large, but the curve in the small frequency region is not a totally straight line. This phenomenon implies the deviation of the capacitive behavior, which may be the main reason for the large IR drop. Furthermore, the Ragone plot was obtained by using the GC/D curves measured using different current densities, as shown in Figure 6e. The maximum energy density of 5.87 Wh/kg was obtained at the power density of 60.00 W/kg. At the maximum power density of 299.93 W/kg, the energy density can still remain 0.27 Wh/kg. Last but not the least, the charge/discharge cycling stability was measured. The relation of the C_M_ retention and the Coulombic efficiency of the FSC to the cycle number in 3000 times charge/discharge cycles were shown in Figure 6f. The C_M_ retention of 70% was remained compared to the C_M_ value at the first cycle in 3000 times repeatedly charge/discharge process. Additionally, the Coulombic efficiency higher than 98% was attained in the whole charge/discharge process. The results indicate that the FSC possesses excellent charge/discharge cycling capability and is able to provide energy in real application continuously. 

The electrochemical performances of the CF@PPy electrode and the fiber-type devices assembled in this work were compared with those achieved in the previous literatures with the similar systems. Table 1 lists the material, the electrolyte for testing the electrochemical performance, and the specific capacitance, as well as the cycling stability for the PPy-based energy storage electrodes and devices reported in the previous works and this work. The CF@PPy electrode prepared in this work shows a competing C_M_ value as those reported in the previous works. The C_M_ value obtained in this work is even higher than that for the electrode incorporated with extra rGO. On the other hand, the electrochemical performance of the FSC prepared in this work was also compared with those reported in the previous literatures. Commonly the carbon materials were incorporated with the conducting polymer to enhance the cycling stability. The specific capacitances were reported based on unit length, unit area or unit weight, so the comparison between different reports is hard to make. Hence, even the C_M_ value of the FSC prepared in this work is smaller than the values reported in some of the literatures, the cycling stability for this device without the carbon material is still acceptable. The specific capacitance and the cycling stability are expected to further enhance with the better assembling technique for fabricating the FSC. There are some methods to improve the performance of this system. Since many parameters for electropolymerization such as electrolyte composition, deposition current and time, and the electropolymerization configuration were optimized in this work, the main issue may lie on the assembly of the fiber-type full device. Other than twisting two fiber electrodes together, different combination ways such as putting two parallel electrodes together can be applied to examine the feasibility. In addition, since the electrodes were combined by using the gel electrolyte, the optimization of the gel electrolyte is another way to improve the performance of this system. The concentration of PVA and H_3_PO_4_, the amount of gel electrolyte, the drying process and even the coating method for gel electrolyte on the electrodes can be studied to further improve the electrochemical performance of this system in the future works.

As a flexible FSC, the bending ability is very important. To examine the bending ability, different bending angles and times were applied on the FSC. The electrochemical performance of the bended FSC was examined using the CV curves at 10 mV/s, as shown in Figure 7. Figure 7a shows the CV curves for the FSC bended with different angles. The shape of the CV curve shows limited distortion even under 180° bending, and the C_M_ retention of 153% was even obtained for the FSC under the bending angle of 180°. On the other hand, Figure 7b shows the CV curves for the FSC bended with different times. The CV curve of the FSC was examined at every 100 times bending. The C_M_ retention of 148% was achieved for the FSC bended with the angle of 180° for 600 times. The C_F_ retention larger than 100% for the FSC bended at 180° for 600 times indicates its excellent bending ability. The increased C_F_ values for the FSC after bending is probably owing to the full diffusion of the gel electrolyte realized by repeatedly bending. It is promising to use this FSC to store energy for the soft electronics especially due to its excellent bending ability. Establishing the FSC is aimed to combine with flexible electronic devices. The photo for the FSC was shown in Figure 8a. The total length is 5 cm and the two fiber electrodes were twisted with each other tightly. The FSC was bended with the angle of 90° (Figure 8b) and 180° (Figure 8c) without distortion. In addition, this FSC can be fixed on cloth (Figure 8d) or used as earring (Figure 8e) and ring (Figure 8f) with the circle shape. With suitable connections with electronic devices, this FSC is expected to be brought with human beings as decorations. The light-weight and conveniently-taken energy storage device proposed in this work is promising to apply in next generation to achieve more comfortable lives for the modern people. 

## 4. Conclusions

This work systematically studied the fabrication of the CF@PPy electrodes using electropolymerization. The supporting electrolyte is necessary to incorporated in the electrolyte to improve the migration of pyrrole monomers. The larger applied current and longer time for electropolymerization lead to more PPy deposited on CF in the forms of larger diameter and denser packing. The CF@PPy electrode fabricated using 0.5 M pyrrole and 0.3 M NaClO_4_ as the electrolyte, the applied current of 40 mA, and the deposited time of 20 min presents the best energy storage performance with the C_M_ value of 308.2 F/g at 5 mV/s, due to the sufficient deposition and suitable packing density of PPy on CF. The assembly of the fibers in the working electrode for electropolymerization was also studied in the novel way. The flexible solid-state FSC fabricated using the optimized CF@PPy electrodes and the PVA/H_3_PO_4_ gel electrolyte shows an energy density of 5.87 Wh/kg at the power density of 60.00 W/kg. The C_M_ retention of 70% and the Coulombic efficiency higher than 98% were also achieved for the FSC in 3000 times repeated charge/discharge process. The bending test also proves the high capability of enduring repeated uses as a flexible device for this FSC.

## Figures and Tables

**Figure 1 nanomaterials-10-00248-f001:**
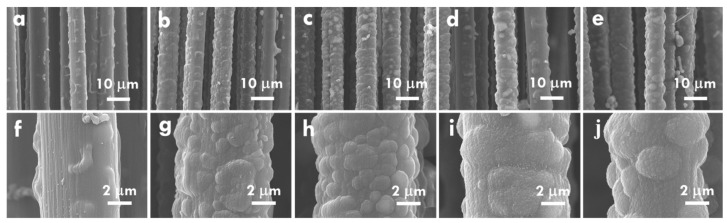
The SEM images for the CF@PPy prepared using the applied currents of (**a**,**f**) 10 mA, (**b**,**g**) 20 mA, (**c**,**h**) 30 mA, (**d**,**i**) 40 mA, and (**e**,**j**) 50 mA for 10 min in the 0.5 M pyrrole and 0.3 M NaClO_4_ electrolyte.

**Figure 2 nanomaterials-10-00248-f002:**
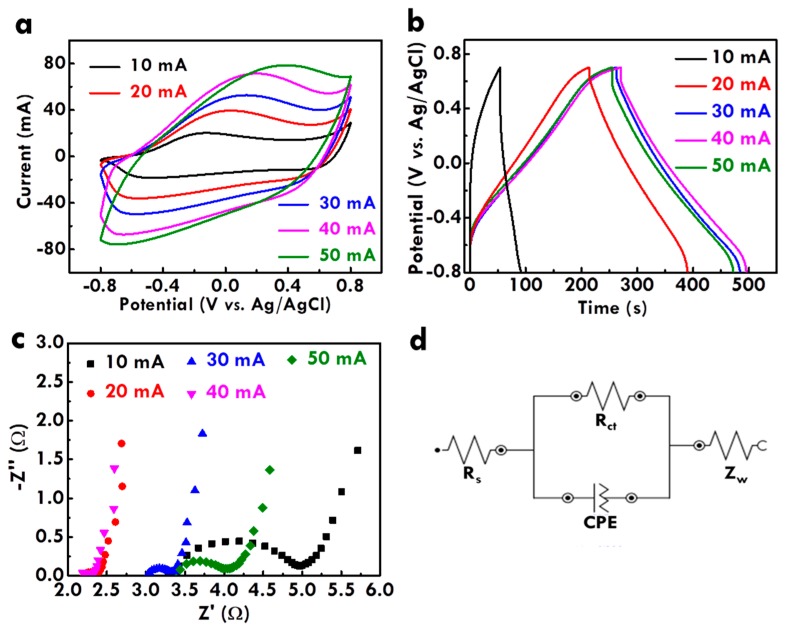
(**a**) The cyclic voltammetry (CV) curves at 20 mV/s, (**b**) the Galvanostatic charge/discharge (GC/D) curves at 1 A/g, and (**c**) the Nyquist plot for carbon fiber on polypyrrole (CF@PPy) prepared using 10 min at different applied currents in the 0.5 M pyrrole and 0.3 M NaClO_4_ electrolyte; (**d**) the equivalent circuit for fitting the Nyquist plot in (**c**).

**Figure 3 nanomaterials-10-00248-f003:**
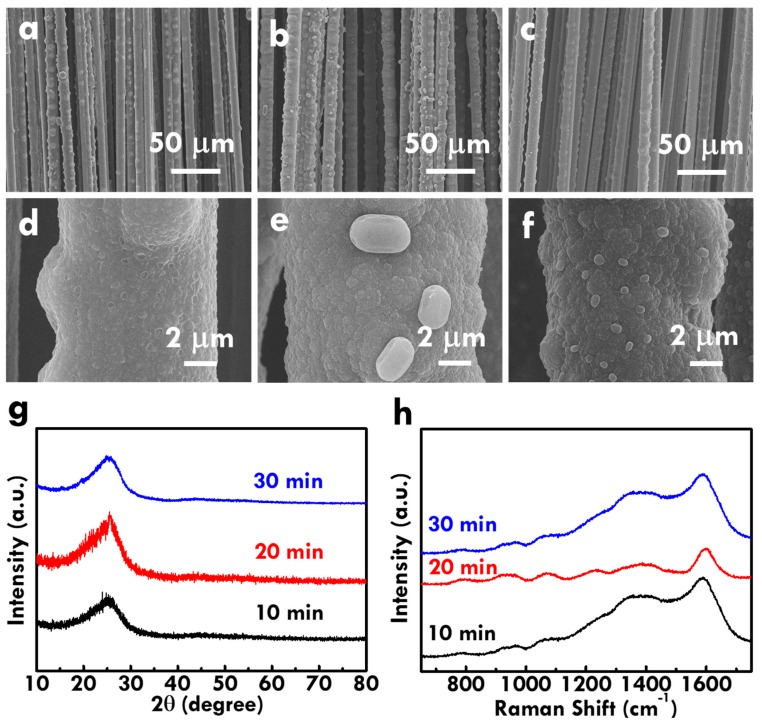
The SEM images for the CF@PPy prepared using electropolymerization times of (**a**,**d**) 10, (**b**,**e**) 20, and (**c**,**f**) 30 min; (**g**) the XRD patterns and (**h**) Raman spectra for the CF@PPy prepared using different electropolymerization times.

**Figure 4 nanomaterials-10-00248-f004:**
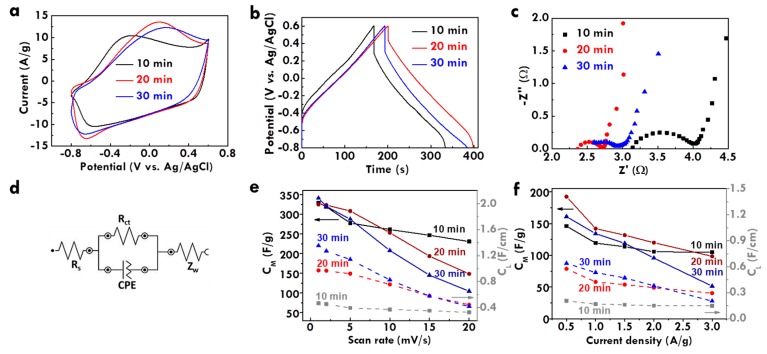
(**a**) CV curves at 5 mV/s, (**b**) GC/D curves at 1 A/g, (**c**) Nyquist plots and (**d**) equivalent circuit for (**c**), (**e**) relation between C_F_ value and scan rate, and (**f**) relation between C_F_ value and current density for CF@PPy electrodes prepared using different electropolymerization times.

**Figure 5 nanomaterials-10-00248-f005:**
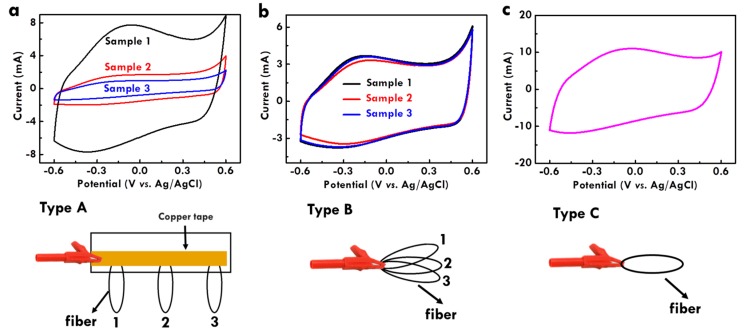
The CV curves for the CF@PPy electrodes prepared using 1 mA and 80 min in the electrolyte of 0.5 M pyrrole and 0.3 M NaClO_4_ with the electropolymerization configurations of (**a**) type A, (**b**) type B, and (**c**) type C.

**Figure 6 nanomaterials-10-00248-f006:**
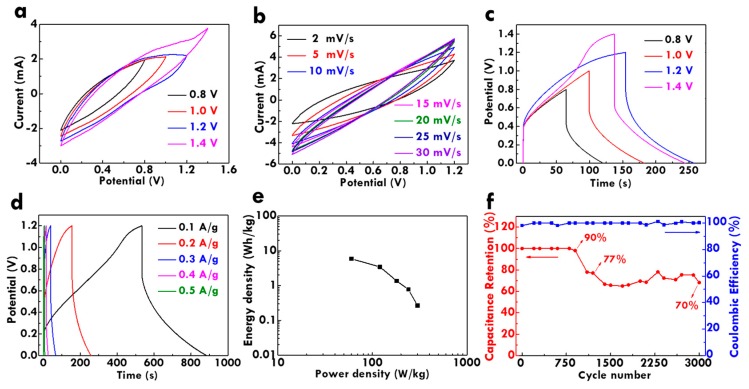
The CV curves (**a**) with different potential windows at 2 mV/s and (**b**) at different scan rates with the potential window of 1.2 V; the GC/D curves (**c**) with different potential windows at 0.2 A/g and (**d**) at different current densities with the potential window of 1.2 V; (**e**) Ragone plots and (**f**) the relation of the C_L_ value and the Coulombic efficiency to the cycle number in 3000 charge/discharge cycles for the fiber-type supercapacitors (FSC) at 0.2 A/g in the H_3_PO_4_/PVA gel electrolyte.

**Figure 7 nanomaterials-10-00248-f007:**
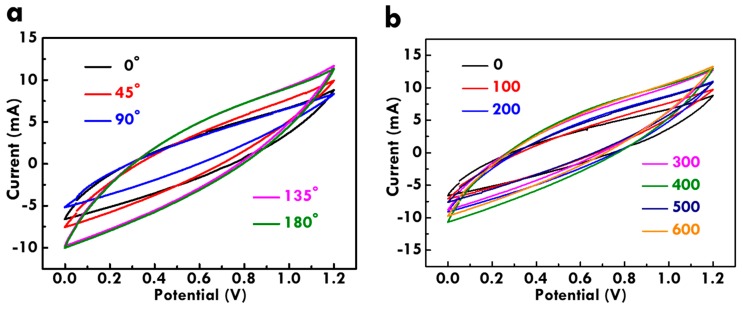
CV curves at 10 mV/s for FSC with varied (**a**) bending angles and (**b**) times at 180°.

**Figure 8 nanomaterials-10-00248-f008:**
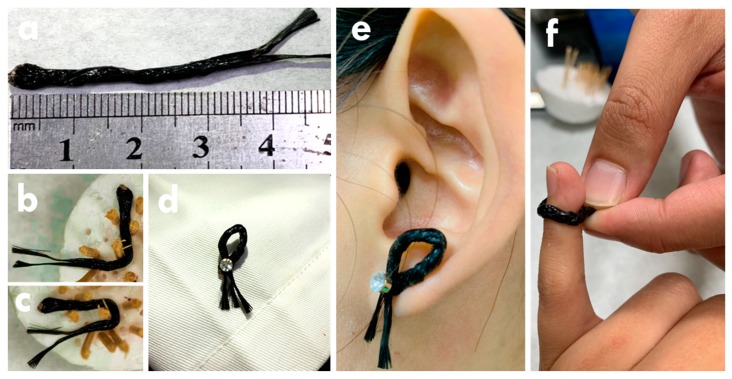
The photos for the FSC (**a**) without bending, (**b**) bended in 90°, (**c**) bended in 180°, bended in the circle shape and (**d**) attached in cloth, (**e**) wore as the earring, and (**f**) wore as the ring.

**Table 1 nanomaterials-10-00248-t001:** Partial list of the material, the electrolyte for testing the performance, and the capacitance as well as the cycling stability for the PPy-based energy storage electrodes and devices.

Materials	Specific Capacitance	Capacitance Retention@Cycle Number	Ref.
**Electrode (Electrolyte = KCl)**			
CF@PPy@rGO	117.5 F/g@25 mV/s	--	[39]
CF@MnO_2_@PPy	315.8 F/g@25 mV/s	82.46%@2000 cycles	[39]
CF@PPy	202.0 F/g@20 mV/s	90%@2500 cycles	[47]
CF@PPy	244.9 F/g ^a^ @20 mV/s	--	TW
**Full Devices (Electrolyte = PVA/H_3_PO_4_)**			
CF@PPy@rGO	188 F/g@2 mV/s	76%@5000 cycles	[47]
CNY@PPy@rGO	111.5 mF/cm@2 mV/s	86%@10000 cycles	[36]
UEF@CNT@PPy	67 mF/cm^2^@5 mV/s	--	[38]
CF@CNT@PPy	3.5 mF/cm@2 mV/s	92%@5000 cycles	[37]
CF@PPy	796.7 mF/g@2 mV/s 30 F/g@0.2 A/g	70%@3000 cycles	TW

CNY = carbon nanofibers yarns; rGO = reduced graphene oxide; CF = carbon fiber; CNT = carbon nanotube; UEF = urethane elastic fiber; TW = This work. ^a^ This value was obtained by using 40 mA as the applied current and 10 min as the applied duration.

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
