# Peer review of "Systematic Design of Polypyrrole/Carbon Fiber Electrodes for Efficient Flexible Fiber-Type Solid-State Supercapacitors"

_nanomaterials, 2020, doi:10.3390/nano10020248_

Round 1

Reviewer 1 Report

After the authors corrections the manuscript is ready for publication

Author Response

General comment

After the authors corrections the manuscript is ready for publication

Response to general comment: Thank the reviewer for the positive comments.

Reviewer 2 Report

After correction manuscript looks much better. However, it reuires some additional work to be acceptable for publications. Here are my comments:

1. Comment  2: Results  of  voltammetric  experiments  obtained  for  [email protected]  should  be compared to these obtained for the same mass of pristine polypyrrole

Response to specific comment 2: Thank the reviewer for the suggestion. However, the voltammetric experiments should be carried out using electrodes. Actually, for the [email protected] electrode, the PPy is the active material and the CF is the substrate. Therefore, it is impossible to measure the voltammetric results for the pristine PPy with CF substrate. We are sorry that we cannot have these results due to the difficulty on measurements.

Reviewer: The same amount of polypyrrole (the same charge of polymerization) should be deposited at the inert electrode. Next, capacitance properties should be measured and compared to the capacitance performance of [email protected]

2. I still have problems with understanding Faradaic impedance measurements. The thickness of [email protected] layer is much lower in comparison to the thickness of solution. Additionally, at potentials of polypyrrole oxidation, polymeric layer is conductive. Therefore, I cannot see any reason of solution resistance dependence on the layer composition.

3. Comment 6: Scheme 1 is very misleading. First of all, the picture is not supported by the experimental data. Polypyrrole is deposited at the positively charged electrode! The electron can be transported by the oxidized polypyrrole layer. The process of counterions incorporation into the polymeric structure is not taken into account.

Response to specific comment 6: Thank the reviewer for the suggestion. We have modified Scheme 1 by replacing electrons with holes in the revised manuscript.

Reviewer: I still think, that there is not enough experimental results to present Scheme 1 even after changes made by Authors of the manuscript.

Author Response

General comments

After correction manuscript looks much better. However, it requires some additional work to be acceptable for publications.

Response to general comment: Thank the reviewer for the second comments.

Specific comments

Comment 1 (Original Comment 2): Results of voltammetric experiments obtained for [email protected] should be compared to these obtained for the same mass of pristine polypyrrole.

Response to specific comment 2: Thank the reviewer for the suggestion. However, the voltammetric experiments should be carried out using electrodes. Actually, for the [email protected] electrode, the PPy is the active material and the CF is the substrate. Therefore, it is impossible to measure the voltammetric results for the pristine PPy with CF substrate. We are sorry that we cannot have these results due to the difficulty on measurements.

Reviewer: The same amount of polypyrrole (the same charge of polymerization) should be deposited at the inert electrode. Next, capacitance properties should be measured and compared to the capacitance performance of [email protected]

Response to specific comment 1: Thank the reviewer for the suggestion. Seems like the reviewer wants to know the contribution of the CF substrate. There are two methods to identify this issue. First, to deposit PPy on inert electrode to eliminate the CF contribution. Second, to carry out the voltammetric experiment for the pure CF substrate. The first method is suggested by the reviewer. However, when the active material being deposited on different substrates, the morphology and the porosity would be different. This phenomenon has been confirmed in our previous work (Electrochim. Acta 2019, 308, 83-90). The effects of morphology and porosity of the active material play very important roles on the electrochemical performance, so the replacement of the substrate from CF to the inert electrode to see the pure electrochemical performance of PPy would not be quite fair to judge the real performance of PPy deposited on the CF substrate as proposed in our work. Therefore, we preferred to use the second method that to compare the electrochemical performance of the [email protected] electrode to the CF electrode. Actually, this data has already been provided in our original manuscript, and the CF substrate shows a very small CF value comparing to that for the [email protected] electrodes. Therefore, it can be highly stated that PPy plays the dominated role on the electrochemical performance in the [email protected] system.

Comment 2: I still have problems with understanding Faradaic impedance measurements. The thickness of [email protected] layer is much lower in comparison to the thickness of solution. Additionally, at potentials of polypyrrole oxidation, polymeric layer is conductive. Therefore, I cannot see any reason of solution resistance dependence on the layer composition.

Response to specific comment 2: Thank the reviewer for the suggestion. The intersection on the real axis of the curve suggests the RS value, which is defined as the series resistance or the solution resistance. Although the reviewer regarded the RS value to be only influenced by the solution, but the composition of the active material really plays visible roles on the RS value. Also, the electrochemical reaction happened at the interface between the active material and the electrolyte, so the active material and the electrolyte may play similar roles on the resistance. In the previous literatures, different RS values were also obtained by using the same solution but different active materials. Peng et al. obtained a significant decrease of charge-transfer resistance for the rGO/PPy/MnO2 aerogel electrode comparing to that of the rGO/MnO2 aerogel electrode, due to the middle PPy coating layer to improve the electron transport from MnO2 to rGO. These systems also show different RS values due to the participation of PPy. (Materials Research Express, 2017, 4, 115602) Xu et al. indicated that the intersection at the real axis represents an internal resistance of supercapacitors, which is relating to the electrical conductivity of electrodes and ionic conductivity of electrolyte. It was suggested that when PPy is applied to electrodes the internal resistance increases due to the degradation of electrical conductivity of electrodes by PPy. (J. Mater. Chem. A, 2015, 3, 22353-22360) Xu et al. found that both the solution resistance and charge transfer resistance of the PPy-fabric are smaller than those of the PPy–RGO-fabric, indicating that the PPy-fabric has better ionic and electronic conductivities perhaps due to the low mass loading on the latter electrode. (Organic Electronics, 2015, 24, 153-159).

Comment 3 (Original Comment 6): Scheme 1 is very misleading. First of all, the picture is not supported by the experimental data. Polypyrrole is deposited at the positively charged electrode! The electron can be transported by the oxidized polypyrrole layer. The process of counterions incorporation into the polymeric structure is not taken into account.

Response to specific comment 6: Thank the reviewer for the suggestion. We have modified Scheme 1 by replacing electrons with holes in the revised manuscript.

Reviewer: I still think, that there is not enough experimental results to present Scheme 1 even after changes made by Authors of the manuscript.

Response to specific comment 3: Thank the reviewer for the suggestion. Then we decided to delete Scheme 1 since this figure did not help to understand the growth of PPy as suggested by the reviewer.

This manuscript is a resubmission of an earlier submission. The following is a list of the peer review reports and author responses from that submission.

Round 1

Reviewer 1 Report

The manuscript "Systematic Design of Polypyrrole/Carbon Fiber Electrodes..." by Sung and Lin presents a series of experiments on the fabrication of fiber supercapacitors where they have varied many common experimental parameters in an effort to optimize the properties of the product FSC. The study is worth publication, but it is neither a high impact or highly novel study. But I believe the data and results are worthy of publication. Some significant issues should be addressed prior to publication.

The general nomenclature for nanomaterials is [email protected] Throughout the manuscript the authors use [email protected] Scheme 1 and throughout the manuscript it is apparent that the designation should be [email protected]  The english throughout the manuscript needs improvement. While it is somewhat readable there are significant issues that make certain sections ambiguous or simply incorrect (as I understand the authors meaning). An example is section 2.1. A sentence says "the pyrrole was coated in CF using the electropolymerization technique." This sentence has, I believe, the opposite meaning of the authors original intent. This is one example of an issue that runs throughout the manuscript. This needs to be fixed. It would be nice to compare the results from the best FSC performance in this current study (maybe a short table) with state-of-the-art performance for these types of systems. The performance for the currently reported systems are not as good as many other similar systems that have been reported (example nanomaterials, (2019) 9, 148, among many others). The authors should indicate whether there are other ways to improve performance for their CF @ PPY systems. Are there other parameters that they plan to look at to improve the performance of this platform.

These issues need to be addressed before the manuscript is ready for publication.

Reviewer 2 Report

The artical is very clear written for readers. The simple idea, not complicated goals of study, well performed experiments and appropriate instrumental analysis supports good level of study.

I will only suggest add EIS analysis for FSC with H3PO4 electrolytes. There is obvious drop of voltage after loading - It should be interesting to know given parameters. 

Reviewer 3 Report

The topic of this manuscript is quite important. Recently, there is a great interest in Flexible supercapacitors. However, the manuscript is very unclearly written and some experimental results are wrongly interpreted. Below are some of my major comments:

The process of carbon fibers modification with polypyrrple should be better described. How was the carbon fibers electrode constructed? Results of voltammetric experiments obtained for [email protected] should be compared to these obtained for the same mass of pristine polypyrrole. I cannot understand the procedure of polypyrrole deposition described on the beginning of Section 3.1. As I can understand, initially, the polymer was deposited in pure water. By the way, water is not an electrolyte. The addition of supporting electrolyte, NaClO4, influences the solution resistance. The diffusion coefficient of pyrrole remains almost constant. What does it means “…the higher applied current is expected to provide more electrons for electropolymerization.” (page 8)? It is very misleading sentence, because pyrrole polymerization occurs by the monomer oxidation. It is obvious that polypyrrole deposition under amperostatic conditions depends on the current density. Chronopotentiometric curves obtained for different current densities should be analyzed in order to explain morphological changes presented in Figure 1. It is very likely that the oxidation of water or supporting electrolyte may influence the process of polypyrrole deposition at higher current densities. Faradaic impedance experiments require more serious analysis. What is the reason of the changes of Rs parameter? The changes observed for Z’-Z” plots obtained for different layers are very random. It is difficult to trust in these results. Scheme 1 is very misleading. First of all, the picture is not supported by the experimental data. Polypyrrole is deposited at the positively charged electrode! The electron can be transported by the oxidized polypyrrole layer. The process of counterions incorporation into the polymeric structure is not taken into account… It is also very difficult to understand the way the results presented in Figure 5 was obtained. Data presented in Table 1 cannot be compared, particularly in the case of solid capacitor device. Data presented in the paper were obtained under amperostatic conditions. The rest of results reported in this part of the table were obtained under CV conditions.

There are major comments . There are many other less important problems with the both data presentation and data interpretation. Therefore, I do not recommend the publication of the manuscript.v